# Exploring the Interaction of Cosmic Rays with Water by Using an Old-Style Detector and Rossi's Method

Marco Arcani [1,*], Domenico Liguori [2] and Andrea Grana [3]

1   Astroparticle Detectors Array Laboratory, GAT Astronomical Center, 21049 Tradate, Italy
2   INFN, National Laboratories of Frascati, Associated Group of Cosenza, IIS Liceo Scientifico "Patrizi", 87062 Cariati, Italy; mim_lig@alice.it
3   Lycée "Ermesinde", L-7590 Mersch, Luxembourg; andrea.grana@lem.lu
*   Correspondence: marco.arcani@astroparticelle.it or info@arcarth.com

**Abstract:** Cosmic ray air showers are a phenomenon that can be observed on Earth when high-energy particles from outer space collide with the Earth's atmosphere. These energetic particles in space are called primary cosmic rays and consist mainly of protons (about 89%), along with nuclei of helium (10%) and heavier nuclei (1%). Particles resulting from interactions in the atmosphere are called secondary cosmic rays. The composition of air showers in the atmosphere can include several high-energy particles such as mesons, electrons, muons, photons, and others, depending on the energy and type of the primary cosmic ray. Other than air, primary cosmic rays can also produce showers of particles when they interact with any type of matter; for instance, particle showers are also produced within the soil of planets without an atmosphere. In the same way, secondary cosmic particles can start showers of tertiary particles in any substance. In the 1930s, Bruno Rossi conducted an experiment to measure the energy loss of secondary cosmic rays passing through thin metal sheets. Surprisingly, he observed that as the thickness of the metal sheets increased, the number of particles emerging from the metal also increased. However, by adding more metal sheets, the number of particles eventually decreased. This was consistent with the expectation that cosmic rays were interacting with the atoms in the metals and losing energy to produce multiple secondary particles. In this paper, we describe a new–old approach for measuring particle showers in water using a cosmic ray telescope and Rossi's method. Our instrument consists of four Geiger–Müller tubes (GMT) arranged to detect muons and particle showers. GMT sensors are highly sensitive devices capable of detecting electrons and gamma rays with energies ranging from a few tens of keV up to several tens of MeV. Since Rossi studied the effects caused by cosmic rays as they pass through metals, we wondered if the same process could also happen in water. We present results from a series of experiments conducted with this instrument, demonstrating its ability to detect and measure particle showers produced by the interaction of cosmic rays in water with good confidence. To the best of our knowledge, this experiment has never been conducted before. Our approach offers a low-cost and easy-to-use alternative to more sophisticated cosmic ray detectors, making it accessible to a wider range of researchers and students.

**Keywords:** cosmic rays; electromagnetic cascades; water; detector; muon; electron; gamma rays; radiation; astrobiology





## 1. Introduction

All high-energy particles that collide with matter generate secondary particles, such as primary cosmic rays, which produce atmospheric particle showers. The energy loss of cosmic particles in the atmosphere can be represented by a general relationship: dE/dX. The energy is progressively lost by the particles through interaction with the atmosphere, especially for leptons. At sea level, the predominant secondary cosmic rays are muons, with a certain component attributed to the so-called electromagnetic cascades (electrons and

gamma rays) and a small percentage of hadrons. Beneath the water's surface, the situation changes significantly, as water has a high stopping power for nucleons and electrons. Neutrons and protons can penetrate only a few millimeters of water thickness before interacting and losing all their energy, while electrons can travel a few centimeters. On the other hand, muons have a high penetration power, and the most energetic ones can reach depths of hundreds of meters. High-energy photons, both atmospheric and nuclear, have a good probability of penetrating water for several tens of centimeters before losing all their energy. So, in water, both electrons (generally intended as positrons and negatrons) and gamma-ray photons can initiate an electromagnetic cascade, as happens in the atmosphere. Therefore, with a detector placed below a tank of water or underwater, it will be possible to measure the progressive absorption of cosmic radiation, mainly represented by muons, as well as the production of radiation represented by electrons and gamma rays. In this research, we focus on measuring the interaction of secondary cosmic rays with water at sea level. Our main aim is to show the development of radiative phenomena like electromagnetic cascades in the initial layers of water due to bremsstrahlung and other interaction processes among particles.

Charged particles from cosmic rays can be detected in several ways, from nuclear emulsion to scintillators and Geiger counters. At high energies, combinations of scintillation and Cherenkov detectors are frequently used. Photons are more difficult to detect directly; they first have to create charged particles in an interaction process in order to be detected. In cosmic ray physics, the "coincidence technique" is widely used in order to distinguish a "cosmic" bullet from background noise (natural ionizing radiation). For this task, two or more stacked sensors (GMT or whatever) and an electronic coincidence circuit are needed. Only when a particle crosses all the sensors does the electronics provide an output, resulting from the almost simultaneous—thus coincident—signals from the sensors. In this scenario, one particle strikes two or more sensors at once. Another use of the coincidence method is to detect particles emitted simultaneously from the same interaction or nucleus, and in this case, at least two distinct particles hit two sensors (almost) at the same time. Coincidence measurements are an important tool in the detection of ionizing radiation for a wide range of applications.

Hans Geiger and Walther Bothe utilized the coincidence approach for the first time in 1924 to demonstrate that Compton scattering causes a recoil electron at the same time with the scattered ray. It was not until the development of electronic circuits at the start of the 1930s that the coincidence technique reached its full potential. In those years, electronics was in its early days and was dominated by thermionic tubes (valves). In 1929, Bothe submitted a paper to the *Zeitschrift für Physik* describing a method for registering simultaneous pulses of two Geiger counters by means of a tetrode vacuum tube. In the same period, Rossi improved Bothe's coincidence circuit by using triodes as multiple switches. In some manner, his circuit was the precursor of the "and" gate that is used in today's computers. In comparison to Bothe's approach, which used a single tetrode vacuum tube and could only detect twofold coincidences, Rossi's threefold coincidence circuit provided a tenfold time resolution improvement. The presence of three or more counters in coincidence, arranged in whatever geometrical configuration, opened new possibilities for investigation. It would soon turn out to be crucial for understanding secondary effects like the interactions between cosmic rays and matter [1]. After the discovery of cosmic ray showers in the atmosphere, Rossi and other scientists started to investigate this phenomenon, performing coincidence experiments with GMT out of line (Figure 1) and overlayed by sheets of metal (he used lead, iron, and aluminum). Some of the more important data on showers and interactions among particles came from these simple devices. Rossi obtained two early results that were quite meaningful for understanding the nature of showers.

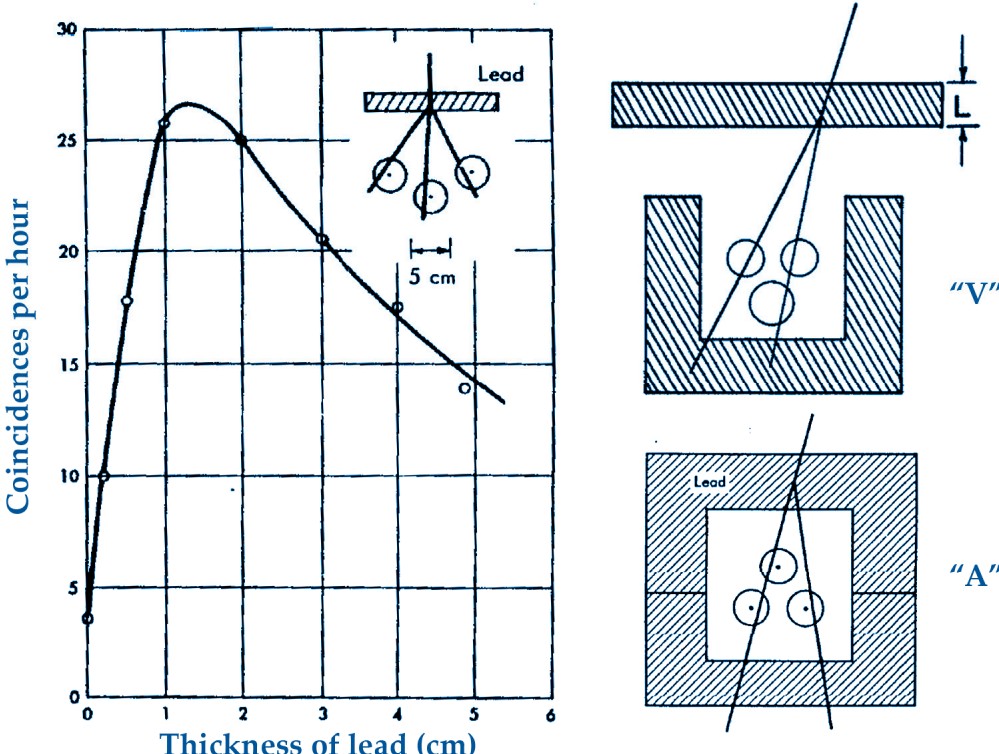

**Figure 1.** Early Rossi's experiment with metals, on the left, the typical shower curve, on the right, different setups for the tubes [2,3].

The first result was the so-called shower curve (thereafter, Rossi's curve), which describes how the coincidence rate changes when the absorber above the counters becomes thicker. The typical shower curve shows that when the material thickness increases, the rate of coincidence due to showers coming out of the metal increases rapidly at first, reaching a maximum between 1 and 2 cm (in lead), and then decreases rapidly again. At the time, the whole picture of cosmic ray physics was very far from complete; hence, this was the proof that cosmic radiation in the atmosphere is composed not only of very penetrating particles, which have a known mean range of the order of meters in lead, but also of radiation responsible for showers much more easily absorbed by lead. Today, we know that the former were muons and the latter were electrons and photons from electromagnetic cascades.

The rate at which showers occurred in various substances was the subject of the second result. Rossi found that by placing, above the counters, layers of lead, iron, and aluminum, all having the same mass per unit area (several grams per square centimeter), there was an approximate shower ratio of 4:2:1 for the three metals. These experimental observations suggested interesting conclusions about the production of showers and led to the development of new theories in particle physics. As we will see later, for a given mass per unit area, radiation losses are much greater in elements of high atomic number than in elements of low atomic number. Despite this, we found and think that particle showers in water should not be neglected in some research fields, such as astrobiology, because of the possible implications for the development of life on Earth and on other planets (see Section 5).

## 2. Theoretical Framework

In cosmic ray physics, the quantities normally used when dealing with the passage of high-energy particles through matter are as follows:

- Interaction mean free path: denoted by a small "λ" (cm), it is the average path of interaction, or else the average distance traveled by a particle between one interaction and another.
- Attenuation length, or radiation length: denoted by a capital "Λ" (or $X_0$) (cm), is defined as the length needed to reduce the energy of a particle to a value 1/e of its original (0.3678, so of 36.78%).
- Interaction depth or radiation depth: denoted by a capital "X", is the path traveled by a particle in a medium and is measured in $gcm^{-2}$ (length times the density of the medium: $cm \cdot g \cdot cm^{-3} = g \cdot cm^{-2}$, from which it can be seen that $\lambda = X/\rho$, thus $g \cdot cm^{-2}/g \cdot cm^{-3} = cm$). Sometimes these quantities may confound because among scientists there is no uniform denotation; some authors express both "λ" and "Λ" multiplied by the density of the medium (in that case λ and Λ are usually denoted, respectively, by λ' and Λ', or by other symbols), becoming likewise $gcm^{-2}$. Even worse, some authors use the lowercase or uppercase Greek letter lambda indifferently.
- Meter of water equivalent (mwe): it is a unit of measurement for attenuation that expresses the thickness of any material as a function of its density in relation to the thickness of a meter of water. This unit is equivalent to the length (thickness) of a medium in meters times its density: mwe = $L_{(medium)}$ [m] ρ [$gcm^{-3}$]. For instance, 0.127 m of iron is equivalent to an mwe (in other words, a meter of water has the same effect as 12.7 cm of iron). The mwe is sometimes convenient, as it makes it possible to directly and intuitively compare the thickness of matter that cosmic rays have to pass through, like in underground laboratories. For example, given that "standard" rock has a density of 2.65 $gcm^{-3}$, a detector placed 380 m below the ground has an attenuation of 1000 mwe, or equal to a column of water measuring one kilometer.

In the atmosphere, λ for primary nuclei with atomic weight A > 25 is quite low and corresponds to about 50 interactions, so there is no chance for a heavy nucleus to reach down to sea level. The attenuation length Λ is different for different kinds of particles and materials; muons have the largest value. In Table 1, we show some values for the electron.

**Table 1.** A few materials with relative radiation length Λ and critical energy Ec for the electron.

| Material | Density ($gcm^{-3}$) | Λ (cm) | $E_c$ (MeV) |
|---|---|---|---|
| Al | 2.7 | 8.9 | 42.7 |
| Fe | 7.87 | 1.76 | 21.7 |
| Pb | 11.4 | 0.56 | 7.4 |
| $H_2O$ | 1 | 36 | 93 |
| Air | $10^{-3}$ [1] | $3.7 \cdot 10^4$ [1] | $\cong 100$ [1] |

[1] At about 1500 m.

In general, the number of particles from an air shower decreases exponentially with increasing atmospheric depth and can be described in a very rough way as:

$$N = N_0 e^{-\frac{d}{\Lambda}}, \text{ also } N = N_0 e^{-\frac{X}{\Lambda'}}, \tag{1}$$

where $d$ is the length in cm, and Λ in cm, while X and Λ' in $gcm^{-2}$.

### 2.1. Photon Intensity Reduction in Matter

When radiative energy loss is dominant, electromagnetic radiation crossing materials undergoes exponential absorption. The intensity is reduced according to:

$$I = I_0 e^{-\alpha d}, \tag{2}$$

where $I_0$ is the initial intensity, $d$ is the distance (cm), and $\alpha$ is the linear coefficient of attenuation (1/cm), which in turn is equal to:

$$\alpha = \mu\rho, \tag{3}$$

with $\mu$ the attenuation mass coefficient ($cm^2g^{-1}$), and $\rho$ the density. So, the Relation (2) can be written as:

$$I = I_0 e^{-\mu d\rho}, \tag{4}$$

And by substituting $d\rho$ with $X$ that is the interaction depth we have:

$$I = I_0 e^{-\mu X}, \tag{5}$$

Many authors alternatively use Equations (2), (4) or (5). These relations can be used as simple models to forecast the behavior of experiments and detectors.

### 2.2. Probability of Interaction

Water and metals are denser than air, resulting in many particles being stopped within the initial layers of these materials. For a particle normally incident on a material and subject to the mean free path process, the probability of interaction is given by:

$$P_{(d)} = 1 - e^{\left(-\frac{d}{\lambda}\right)} \cong \frac{d}{\lambda}, \tag{6}$$

This means that, for instance, 63.2% of all particles will have a collision within $\lambda$ length in a medium with $d$ thickness. The approximation shows that the probability of interaction is simply given by the ratio of material thickness and the mean free path [4].

### 2.3. Development of Electromagnetic Showers in Matter

In this chapter, we discuss some processes related to the particles that can initiate electromagnetic cascades in water. As seen so far, near the water surface of seas and lakes, there are essentially two types of secondary cosmic radiation: hard and soft. By definition, the hard component of cosmic rays is able to penetrate 15 cm of lead [5] (167 g $cm^{-2}$, or 1.67 mwe). Showers of electromagnetic particles can be initiated in the first layers of water (say, in the top 100 cm) by a few penetrating particles, namely, muons, electrons, and gamma rays, while more in-depth, only hard components (muons) could start a shower. Muons traveling in water can lose energy through a number of different processes, including ionization and excitation, direct electron pair production, bremsstrahlung, and photo-nuclear interactions. The total energy loss can be described as:

$$-\frac{dE}{dx} = a(E) + b(E)E, \tag{7}$$

where the term $E$ is the total energy, $a(E)$ is the electronic stopping power (ionization and atomic excitation), and $b(E)$ is a composite term due to radiative processes that are bremsstrahlung, pair production, and photonuclear interactions:

$$b(E) = b_{br}(E) + b_{pp}(E) + b_{ni}(E), \tag{8}$$

Muon decay, other than neutrinos, can leave high-energy electrons (despite a very low probability, some muons can also decay into the experiment):

$$\mu^+ \rightarrow e^+ + \nu_e + \bar{\nu}_\mu \; \mu^- \rightarrow e^- + \bar{\nu}_e + \nu_\mu \tag{9}$$

High-energy electrons lose the majority of their energy via bremsstrahlung radiation. As a result, in their interactions with matter such as water, the majority of the energy is spent creating high-energy photons:

$$e^\pm \to e^\pm + \gamma, \tag{10}$$

while only a small portion is lost in other processes.

Atmospheric photons of cosmic origin arise from a great variety of processes. The processes involved in the production of gamma and X-rays include nuclear and electromagnetic interactions such as neutral pion decay, various decays of other unstable particles, bremsstrahlung of electrons and positrons, positron–electron annihilation, and the production of Cherenkov radiation. The following list presents some examples of energy levels and the corresponding phenomena associated with photons in the atmosphere. From higher to lower energy levels, each entry describes different processes and interactions [6]:

1. The decay of neutral pions results in the emission of two gamma rays with a combined energy of at least 140 MeV;
2. Electrons from muon decay and other accelerated electrons have a 30–40% chance of producing bremsstrahlung;
3. The annihilations between electrons and positrons produce a characteristic energy line at 0.51 MeV;
4. Nuclear collisions yield neutrons with energies of approximately 10 MeV; these neutrons can undergo scattering or be captured by nitrogen-14 and oxygen-16, resulting in the creation of excited states that emit characteristic gamma energies;
5. Gamma rays below 2 MeV degrade slowly by multiple Compton scattering;
6. Photons around 30 keV have interactions through the photoelectric effect.

At ground level, there are no data for only photons. All measurements include electrons and gamma rays combined. The ratio of photons to electrons above 100 MeV was found to be unity (1:1), and the intensity is in the order of $10^{-3}$ cm$^{-2}$s$^{-1}$sr$^{-1}$ (Palmatier 1952, Greisen 1942, Chou 1953) [7]. High energy photons can trigger off the pair-production in metals and water:

$$\gamma \to e^+ e^-, \tag{11}$$

However, pair production can occur only above the threshold energy of two-electron masses:

$$E_{th} = 2m_e c^2 = 2 \cdot 511 \text{ keV} = 1.02 \text{ MeV}, \tag{12}$$

These secondary photons, in turn, undergo pair production; then, electrons and positrons can in turn radiate. According to the available energy, this phenomenon continues to generate particles that are called electromagnetic showers, electromagnetic cascades, or even electrophotonic showers (cascades). A simple model of an electromagnetic cascade is called the Heitler's toy model (by Walter Heinrich Heitler, Figure 2).

In a cascade, the number of branches $n$ at a distance $d$ doubles at every radiation length where

$$n = \frac{d}{\Lambda}, \tag{13}$$

After crossing $n$ radiation lengths $\Lambda$, the total shower size is:

$$N = 2^n = e^{\frac{d}{\Lambda}}. \tag{14}$$

Considering a shower in water initiated by a single particle with energy $E_0$, the cascade continues until it reaches a maximum number of particles. When the energy of the particles falls below the critical energy $E_c$ (i.e., radiative energy loss is equal to ionization energy loss),

the shower is halted because bremsstrahlung no longer dominates, so that the maximum number of particles is defined as:

$$N_{max} = \frac{E_0}{E_c},\tag{15}$$

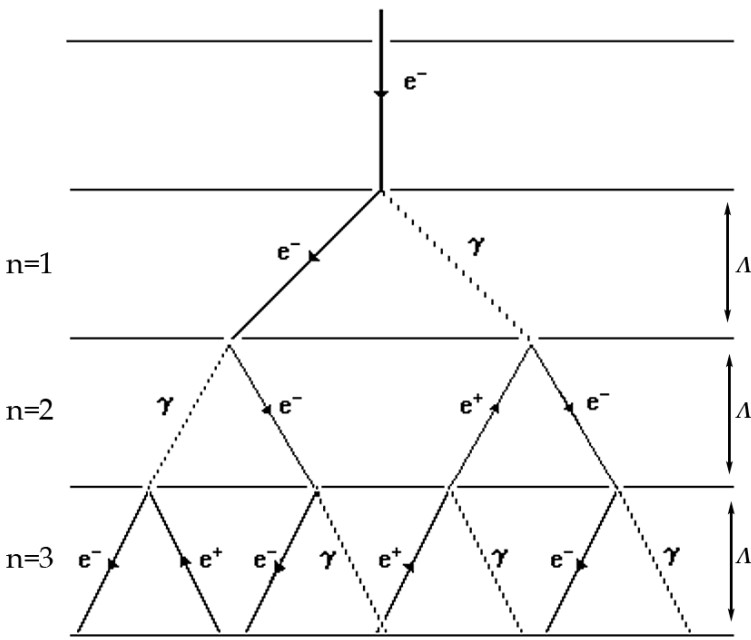

**Figure 2.** An electromagnetic shower initiated by an electron.

The maximum depth $d_{max}$ at which the cascade reaches its maximum size is given by an equation derived from this model:

$$d_{max} = \Lambda \, \frac{ln\left(\frac{E_0}{E_c}\right)}{ln2} \; (a), \quad d_{max} = \Lambda \, ln\left(\frac{E_0}{E_c}\right) - 0.5 \; (b), \quad d_{max} = \Lambda \, ln\left(\frac{E_0}{E_c}\right) \; (c),\tag{16}$$

where ($a$) is the original model by Heitler, for ($b$) and ($c$), see [4,8,9] in the bibliography. Given that $d_{max}$ describes the depth of the cascade maximum in terms of radiation length, and since $\Lambda$ is proportional to $A/Z^2\rho$, it follows that electromagnetic showers are deeper the lower the atomic number and the lower the material density.

As shown in Figure 2, the total radiation depth $X_{max}$ crossed by a shower is a function of the radiation length traveled ($\Lambda$), its number, and the density of the medium, as follows:

$$X_{max} = n_c\Lambda\rho = n\Lambda\iota.\tag{17}$$

This highlights that the number of radiation lengths is proportional to the density of the medium, and since the number of particles doubles with every radiation length, the total number of particles generated is also proportional to the atomic number of the element being traversed (given that $\Lambda$ is proportional to $A/Z^2\rho$). However, if the thickness is measured in units of interaction length, the development of the shower is independent of the material.

In summary, in the experiment, we expect that electromagnetic cascades of particles can be initiated by a single electron, gamma ray, or muon with high energy ($\cong$GeV) in different layers of water. Increasing the thickness of water above the detector should increase the chance of producing more showers, in the same fashion as it happens with metals. Our instrument can detect the showers mainly in the form of three coincident particles, at least.

## 3. Experimental Setup

The instrument was built as part of our ADA cosmic ray project [10]. This device has been named AMD16 (we nickname all the devices "Astroparticle Muon Detector" followed by a model number) and was designed specifically to detect showers and eventually to remake a pioneering experiment made by Domenico Pacini to investigate cosmic radiation beneath the water's surface [11].

### 3.1. The Cosmic Ray Telescope and Shower Detector AMD16

The main components of the detector are four GMTs, an electronic board, and an Arduino data logger. Two GMTs have a vertical alignment and are intended for muon detection with the coincidence method; while the other two GMTs, out of line and both in coincidence with one of the previous, are intended for shower detection. They are oriented in such a way that no single ionizing particle can travel across all three simultaneously. In more detail, GMT-1 and GMT-2 are two metal Geiger tubes model SBM-19 (Soviet production), while GMT-2 and GMT-3 are two glass Geiger tubes model J-306 (Chinese production). Their dimensions are almost identical (195 mm × Ø18 mm), but J-306 seems to be more sensitive and "noisy". SBM-19 tubes have been tested for many years and are also used in big equipment, like the CARPET detector [12]. Unfortunately, the main suppliers are Ukrainians, and the current situation hampers obtaining these sensors. GMT-2 and GMT-3 are separated by 7 cm, whereas GMT-2 and GMT-3 are separated by 5 cm. Figure 3 shows the arrangement of the four GMTs.

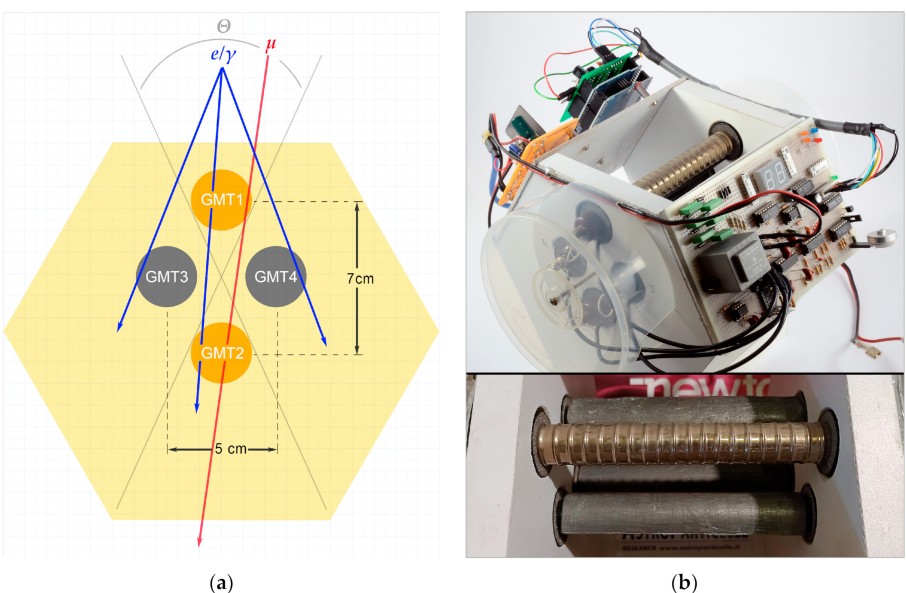

**Figure 3.** Instrument AMD16: (**a**) Cross-section of the detector; (**b**) An image, at the bottom a close-up of the GMTs, in this case, three of them shielded by lead.

When a particle crosses a tube, a signal with a pulse length of about 190 μs is generated (dead time + recovery time). Every signal is then enlarged to 550 μs by a monostable oscillator that works as a trigger for the coincidence; this time window is the maximum resolving time of the detector. Signals from GMT-1 and GMT-2 are combined in an "and" gate; if they arrive within the trigger time window, this signal is accounted for as a muon and sent to the data logger for counting and recording. In the same fashion, signals generated in GMT-1, 3, and 4 are combined in a second "and" gate and recorded as a shower. The propagation delay time of the integrated circuit ("and" gates) is on the order of 100 ns. This represents the minimum time for two overlapping signals (minimum resolving time); thus, to have a coincidence, the input signals must arrive within the range 0.1 ÷ 550 μs (Figure 4). The circuit does not require an extremely fast resolution time because all of the counting rates are low.

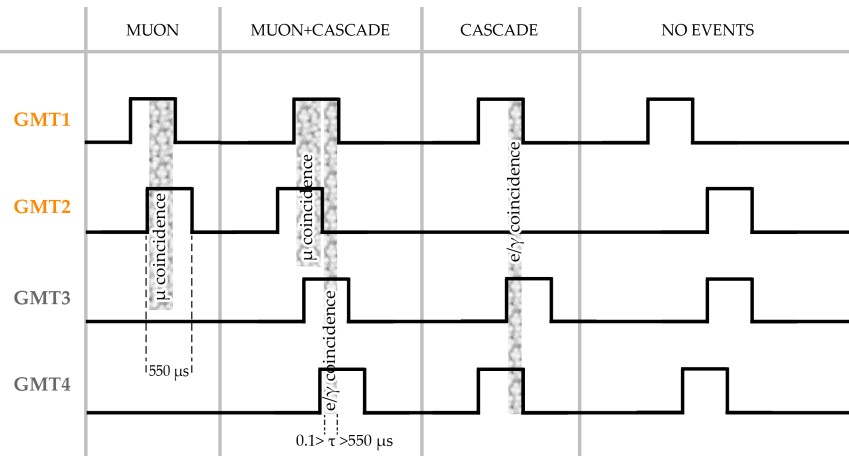

**Figure 4.** Examples of several possible combinations of signals in the detector.

While in the muon event, two GMTs "blip" for the passage of one single particle; in the shower event, the three GMTs "blip" for the passage of at least three particles because, being out of line, no single particle can cross all three. Notice that Rossi used both a kind of "V" and "A" configuration in his experiments (maybe he would have used GMT-2, 3, and 4; see also Figure 1), while our setup is like an "A" configuration. Anyway, it is the same thing, since when a shower event occurs, GMT-1 and GMT-2 are always crossed by some particles. This was confirmed by means of an oscilloscope and a video camera, which shot the LEDs related to each GMT event; thus, in actuality, in a shower event, all four GMTs are involved. The detector in total has six independent channels: Muon, Shower, GMT-1, GMT-2, GMT-3, and GMT-4. These signals from the main electronic board are sent to the data logger through a buffer; the outputs from the buffer can also be connected to other interface boards for connection to personal computers. The main board also has a seven-segment LED display counter to track muons or shower events. The presence of a buzzer that clicks when whatever particle crosses a sensor is useful to check the correct working state of the detector. A block diagram of the device is shown in Figure 5.

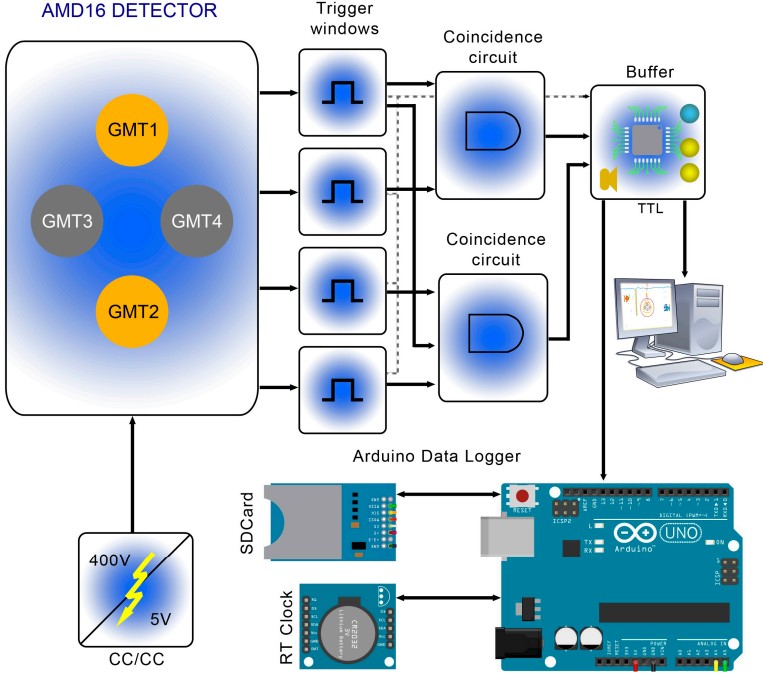

**Figure 5.** Simplified block diagram of the detector AMD16.

### 3.2. Geometry of the Muon Telescope and Shower Detector

A geometry problem that usually arises in particle detection is the calculation of the solid angle subtended by a single detector. This will often involve numerical integration starting from the solid angle definition:

$$\omega = \iint \sin\theta\, d\theta\, d\varphi, \tag{18}$$

For a detector with two or more aligned sensors (GMTs), the calculation cannot be trivial. The effective muon solid angle depends on both the size and distance of two sensors, as well as the detection surface and its shape. There are several ways to achieve the value of the solid angle; however, the geometry can be calculated assuming some simplifications. First, we consider the two cylindrical sensors as two 2D rectangles, and then we calculate the angles $\theta$ and $\phi$. Recalling that for a muon to make a coincidence between the top and bottom sensors, it must pass through both; those angles would be along the path from one edge of the top sensor to the opposite edge of the bottom sensor. In this manner, we obtain a single rectangle whose projection on the unit sphere is the solid angle of the whole detector (Figure 6).

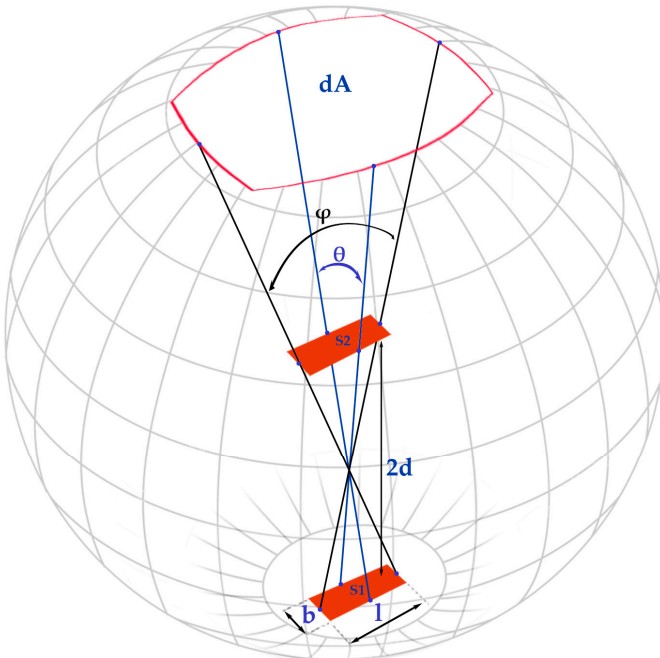

**Figure 6.** Conceptual geometry of the muon telescope of the detector (not to scale).

Instead of dealing with angles and differentials, for the calculation we used an interesting and simpler method by H. C. Rajpoot using the following Equation [13]:

$$\omega = 4\sin^{-1}\left(\frac{lb}{\sqrt{\left(l^2 + 4d^2\right)\left(b^2 + 4d^2\right)}}\right), \tag{19}$$

where "$l$" and "$b$" are the sides of the rectangle and "$d$" is the semi-distance of the two sensors.

It comes out that the solid angle of the cosmic ray telescope (muon detection) is close to a unit steradian: 0.92 sr. Being the integral intensity of vertical muons above 1 GeV/$c^2$ at sea level $0.7 \cdot 10^{-2}$ (cm$^{-2}$s$^{-1}$sr$^{-1}$) (value taken from Particle Data Group Booklet, PDG 2022), the theoretical acceptance of the telescope for muons should be about 0.38 (cm$^{-2}$min$^{-1}$). Since the detector averages a flux rate of about 0.18 (cm$^{-2}$min$^{-1}$) (about five counts per minute, or cpm for about 27 cm$^2$), its efficiency for muon counting (considering the ratio between

the actual counts and the theoretical counts) should be approximately 0.18/0.38 = 0.47, i.e., 47%.

For shower detection, the same calculations are very complicated and can be performed using the Monte Carlo method, but to date, we have no such data. The complications arise because the three (or four) sensors in coincidence separately see a solid angle of $4\pi$ sr each, or $2\pi$ sr considering only the upper half of a hemisphere, but the acceptance also depends on the geometric shape for the coincidence and other factors.

Electrons are detected by a direct mechanism since they directly induce ionization by collision with the gas inside the tubes. Low-energy photons are also detected in a kind of direct process due to photoionization. High-energy photons instead are detected via an indirect process; indeed, high-energy photons eject electrons at the inner surface of the tube's cathode by photoemission, and those electrons ionize the gas. So, the efficiency of shower detection is related to the efficiency of the GMTs and is near 100% for electrons, while for gamma radiation it is proportional to the energy of photons. In the energy range of 30 keV to 1.25 MeV, photon radiation detection efficiency is as low as a few percent, allowing the majority of radiation to pass the tubes without ionizing them [14]. The counting efficiency for gamma rays depends on the probability of a gamma ray interacting with the cathode wall and producing an electron, as well as the probability that the electron will ionize the detection gas before losing all its energy.

### 3.3. Methodology for Detecting and Measuring Particle Showers in Water

For the setup of experiments, there were two main reasons for concern: the first related to the use of tap water, and the second related to the use of GMT shields. Tap water is rich in minerals and other substances that can potentially interfere with shower production, so using distilled water could be a better choice. Some particles can also bounce from one tube to another, altering the results. Shielding the GMTs with thin lead foils can prevent this "recoil" effect; by the way, this also cuts off low-energy particles from the shower itself. In order to evaluate these (and other) problems, we performed several trials. The very first test was a measurement under the Rhone Glacier at Furka Pass in Switzerland. The detector was placed in the "Ice Grotto", an artificial cave at the front of the glacier that is re-drilled each year. This was useful to assess the absorption of muons by the ice, but unfortunately, the ice sheet above the tunnel is several meters thick, too much to assess shower particles (Figure 7).

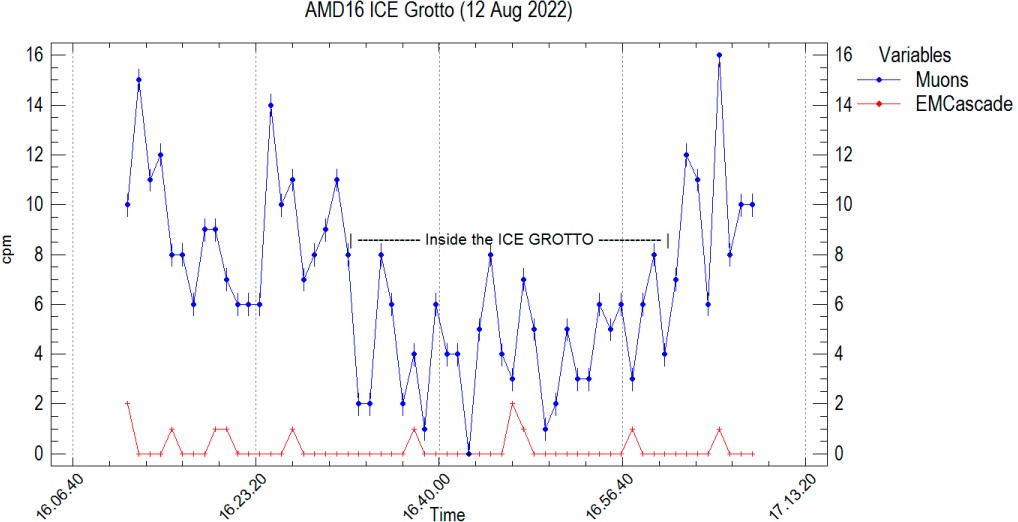

**Figure 7.** First test of the behavior of the detector under the glacier's ice.

In the laboratory, we made two main measurements: a measurement under a tank of water up to 100 gcm$^{-2}$ and a measurement under plates of iron up to about 118 gcm$^{-2}$. The first trial was made both with tap water and distilled water up to 45 gcm$^{-2}$, but since the results were quite similar between the two types of water, the test continued only with tap water, and all the data are related to it. Further, from 0 to 100 gcm$^{-2}$, we made two distinct measurements, with and without GMT shielding, and in this case, the results were pretty much different.

## 4. Results and Analysis

In coincidence measurement, it is important to know the number of accidental coincidences to be expected among the sensors. The term "accidental" refers to the occurrence of simultaneous events in multiple detector channels that are not caused by the particles of interest. These accidental counts can arise due to various sources of background radiation or noise, and they can affect the accuracy and reliability of the particle detection system. If $m$ is the mean value of particles counted by a GMT and $n$ is the number of GMTs, the general relation can be expressed as [15]:

$$A_n = n \cdot \overline{m_1} \cdot \overline{m_2} \cdot \ldots \overline{m_n} \cdot \tau^{n-1} \tag{20}$$

where $\tau$ is the resolving time of the detector. Usually, the greater the number of tubes in coincidence and the shorter the resolving time, the fewer the accidental coincidences. In our case, the expected number of accidental coincidences was 0.045 counts per hour (cph). However, this result is not consistent with our measurement; we will tackle the cause of this issue in the discussion.

### 4.1. Electromagnetic Cascades in Iron

In order to repeat Rossi's experiment, we used 15 plates of iron big enough to cover the top detector's surface (10 × 17 cm) and 1 cm thick; the distance from the top of the detector to the first slab of iron was no more than 2.5 cm. We used iron instead of lead because it is easier to find among suppliers. We gathered 15 sets of data from 1 to 15 cm thickness, plus a background measurement with no metal above the detector; every measurement lasted not less than 8 h. The detector provides a count every minute, so we had more than 480 counts for every measure. Thus, the data were integrated to count per hour, to avoid too many decimals, and to have a direct comparison with Rossi's data. For every measurement, the standard error was around 5%, as shown by the error bars in Figure 8. Then the experiment was repeated in the same manner, but this time with a shield of lead (2.5 mm thick) around the three lower GMT. The results are shown in the graphs in Figure 8.

In both cases, there are some interesting results; the former is the first higher peak of the curve at around 20 gcm$^{-2}$, with another prominent peak at around 70 gcm$^{-2}$ in Figure 8a and at 86 gcm$^{-2}$ in Figure 8b. The first peak is expected to be the maximum of the shower and corresponds exactly to 23.62 gcm$^{-2}$ or 3 cm of iron (in both cases); this can be compared to the early results of Rossi (Figure 1), where he obtained the maximum in correspondence to 1.6 cm of lead, which from Equation (17) is equivalent to 18.4 gcm$^{-2}$. The expected Rossi curve is better represented by the fit of the first run, where the tubes were not shielded. The second big maximum is a controversial feature already seen in the literature known as "the second maximum of the shower transition curve" [16]. As introduced before, the "background noise" is relatively high: 16.7 cph on the first run vs. 13.4 cph on the second run. In the second graph, the lower rate due to the lead shield is evident, but despite the shielding, the background still remains high.

Electromagnetic Cascade (Fe)

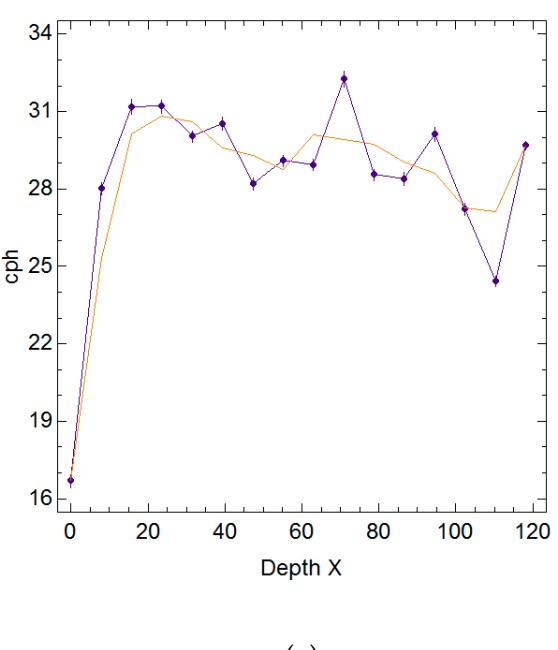

Electromagnetic cascade (Fe GMT shielded)

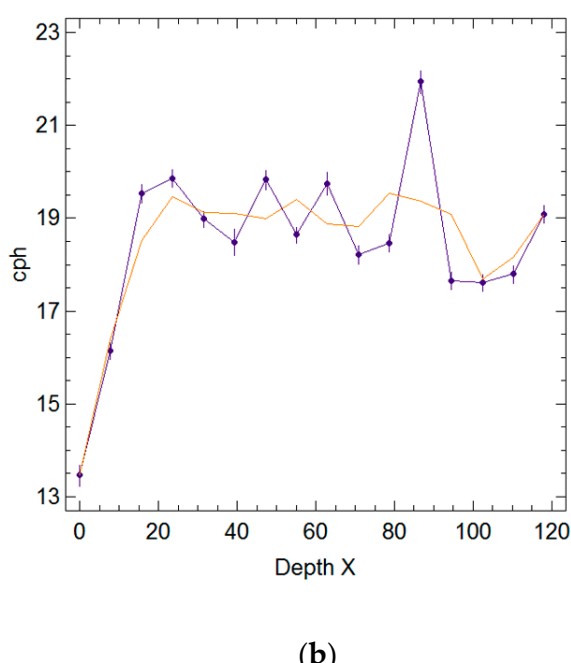

(**a**)

(**b**)

**Figure 8.** Measurement of electromagnetic cascades in iron from 0 to 15 cm; the value on the abscise is expressed in radiation depth ($gcm^{-2}$); the orange line is a kind of fit represented by the running mean: (**a**) First run with GMT unshielded; (**b**) Second run with GMT shielded by lead.

### 4.2. Electromagnetic Cascades in Water

In the beginning, we made two runs for this measurement, both with distilled water and tap water, in order to assess any difference between them. The data did not show any substantial difference in the trend, despite the fact that in distilled water the count rate seems to be slightly higher (Figure 9). Cosmic ray experiments are sensitive to variations in the general cosmic radiation trend due to weather conditions (atmospheric pressure and temperature) or solar activity, so the difference between the two sets of data is mainly caused by those effects.

Comparison of cascades in different type of water

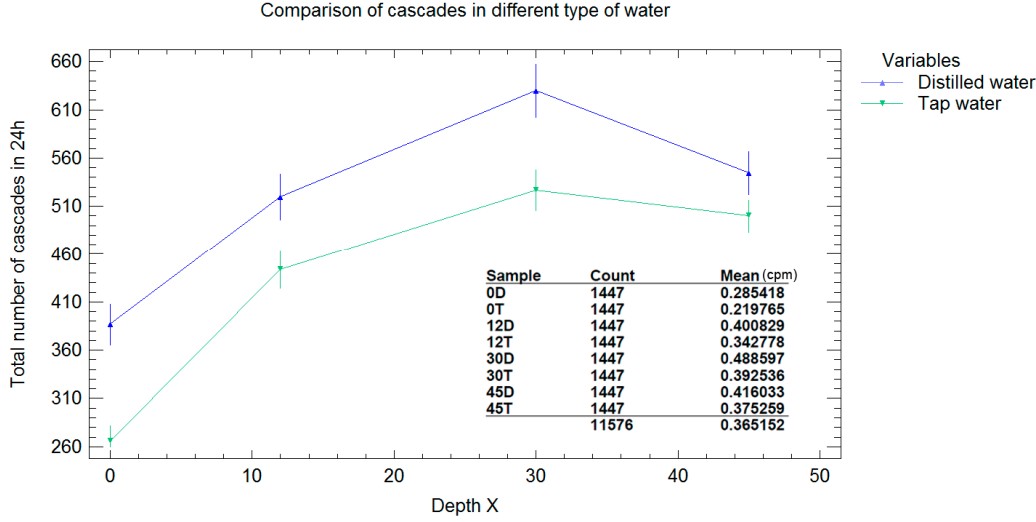

**Figure 9.** Comparison of data between the two types of water.

The main two runs followed a similar procedure as the experiment with iron plates, shielding the GMT in the second run. To hold on to the heavy load of water, we used interwoven wooden beams; thus, the distance from the top of the detector to the bottom of the tank was about 10 cm. In these experiments, we gathered a total of seven sets of data for every trial at this step: 0, 12, 30, 45, 60, 75, and 100 $\text{gcm}^{-2}$ of water above the detector; every measurement lasted at least 24 h. As for the experiments with iron, the detector provides a count every minute, so we had more than 1447 counts for every measure. Then, the data were integrated to calculate the count per hour. The results are shown in the graphs in Figure 10. For every measurement, the standard error was around 5%, as shown by the error bars and from the analysis in Table 2.

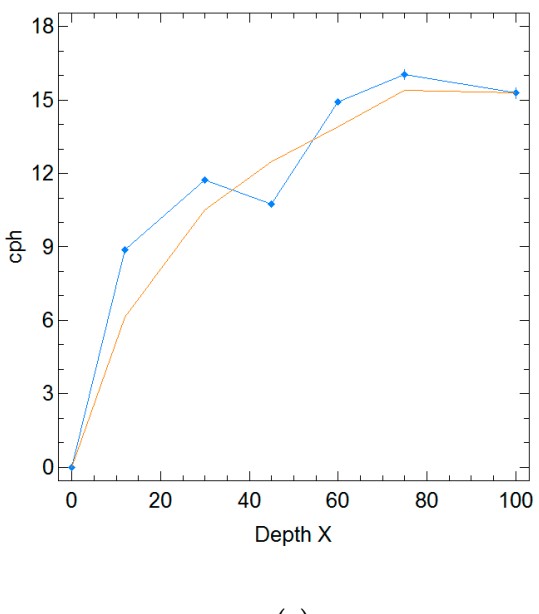

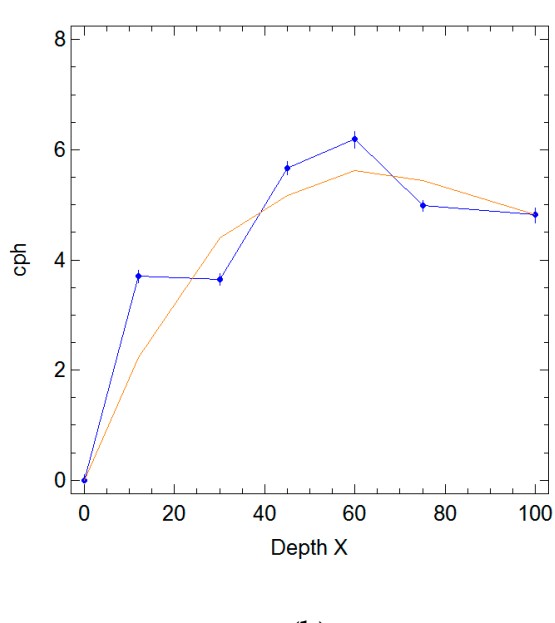

(**a**) (**b**)

**Figure 10.** Measurement of electromagnetic cascades in water from 0 to 100 cm; the value on the abscise is expressed in radiation depth ($\text{gcm}^{-2}$); the orange line is a kind of fit represented by the running mean: (**a**) First run with GMT unshielded; (**b**) Second run with GMT shielded by lead.

**Table 2.** The table shows 95.0% confidence intervals for the means and standard deviations (counts per minute) of each of the variables in the first run (1) and in the second run with GMT shielded by lead (2).

| Water X ($\text{gcm}^{-2}$) | Mean (cpm) | | Stnd. Error | | Lower Limit | | Upper Limit | |
|---|---|---|---|---|---|---|---|---|
| | **(1)** | **(2)** | **(1)** | **(2)** | **(1)** | **(2)** | **(1)** | **(2)** |
| 0 | 0.20 | 0.22 | 0.012 | 0.016 | 0.17 | 0.19 | 0.22 | 0.25 |
| 12 | 0.35 | 0.28 | 0.015 | 0.015 | 0.32 | 0.25 | 0.38 | 0.31 |
| 30 | 0.40 | 0.28 | 0.016 | 0.014 | 0.36 | 0.25 | 0.43 | 0.31 |
| 45 | 0.38 | 0.31 | 0.013 | 0.016 | 0.35 | 0.28 | 0.40 | 0.35 |
| 60 | 0.45 | 0.32 | 0.018 | 0.020 | 0.41 | 0.28 | 0.48 | 0.36 |
| 75 | 0.47 | 0.30 | 0.030 | 0.014 | 0.41 | 0.27 | 0.53 | 0.33 |
| 100 | 0.45 | 0.30 | 0.030 | 0.018 | 0.39 | 0.27 | 0.51 | 0.34 |

In this case, we have subtracted the background to give more clean graphs; the results are quite good, and the effect of electromagnetic cascades in water is clear. The maximum of the showers appears at 75 gcm$^{-2}$ for the first run and at 60 gcm$^{-2}$ for the run with shielded GMTs. In both graphs, there is a flatness or inflection of the curve above, say, 80 gcm$^{-2}$. This means that over this depth, the particles are going to be halted, losing all their energy, and the maximum of the showers is surely reached between 60 and 75 gcm$^{-2}$. In these experiments, the effect of the lead shield is still evident; in the second run, the average count is about half that of the first run, but anyway, the trends are very similar. We used the models of Equation (16) to evaluate the energy of the particle necessary to produce a shower reaching the maximum depth; the results are shown in Figure 11 and in Table 3.

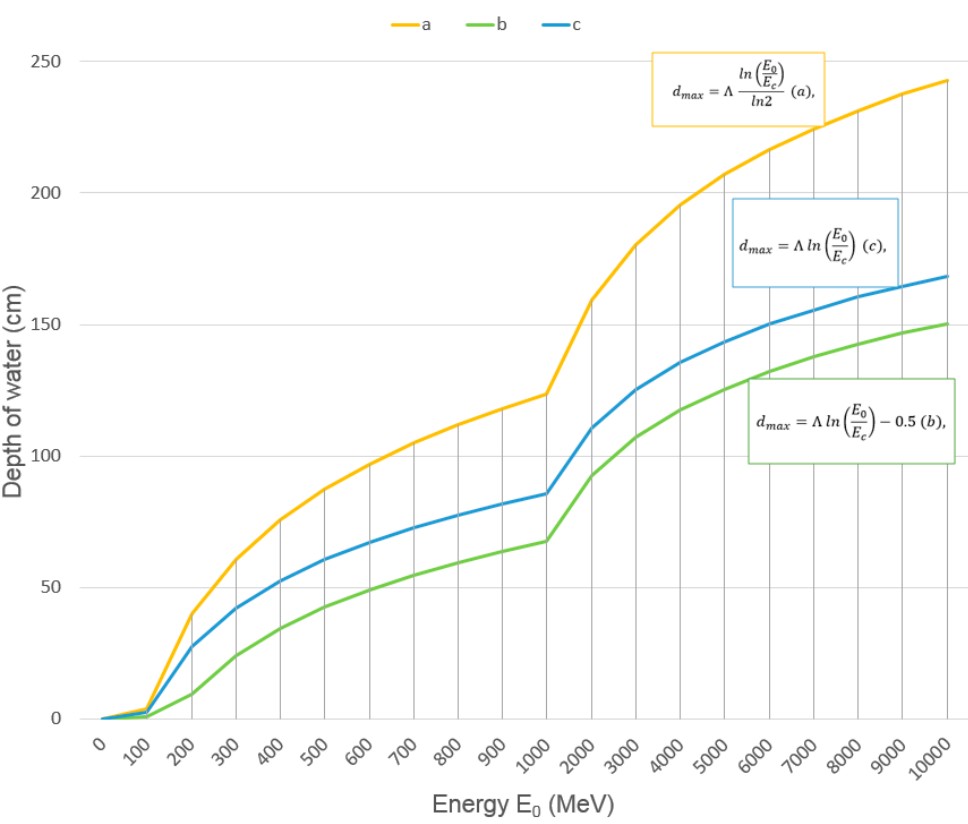

**Figure 11.** Graph of the models from Equation (16).

**Table 3.** Approximate evaluation of the energy $E_0$ of the particle that initiated the cascade.

| Element | $E_0$ (Equation (16a)) (MeV) | $E_0$ (Equation (16b)) (MeV) | $E_0$ (Equation (16c)) (MeV) |
|---|---|---|---|
| Pb | 60 | 200 | 150 |
| Fe | 80 | 200 | 150 |
| H$_2$O * | 300 | 900 | 500 |
| | 400 | 1000 | 750 |

* For the range 60÷75 cm of depth.

Using Equation (15), it is also possible to evaluate the number of maximum particles generated in a cascade. Experimental data from other authors show that the original model by Heitler needs some correction to fit the experimental data, so we trust more in the other two tested models. From these two, it is found that the energy ($E_0$) for the initial particle should be in the range $0.5 \div 1$ GeV (Table 3). Now, using Equation (15), we can estimate that the number of particles $N_{max}$ should be from 4 to 12. This is consistent with Equation (13), which forecasts a value of two for *n*, meaning that the number of photons and electrons from a shower in 72 cm of water initiated by a single particle should not be more than six. More precisely, if an electron initiates the cascade, to produce six particles its initial energy should be approximately 558 MeV.

### 4.3. Muon Absorption in Water and Iron

This is not the main topic of the paper, but for the sake of completeness, we also show the response of the instrument to muon detection through iron and water (Figure 12).

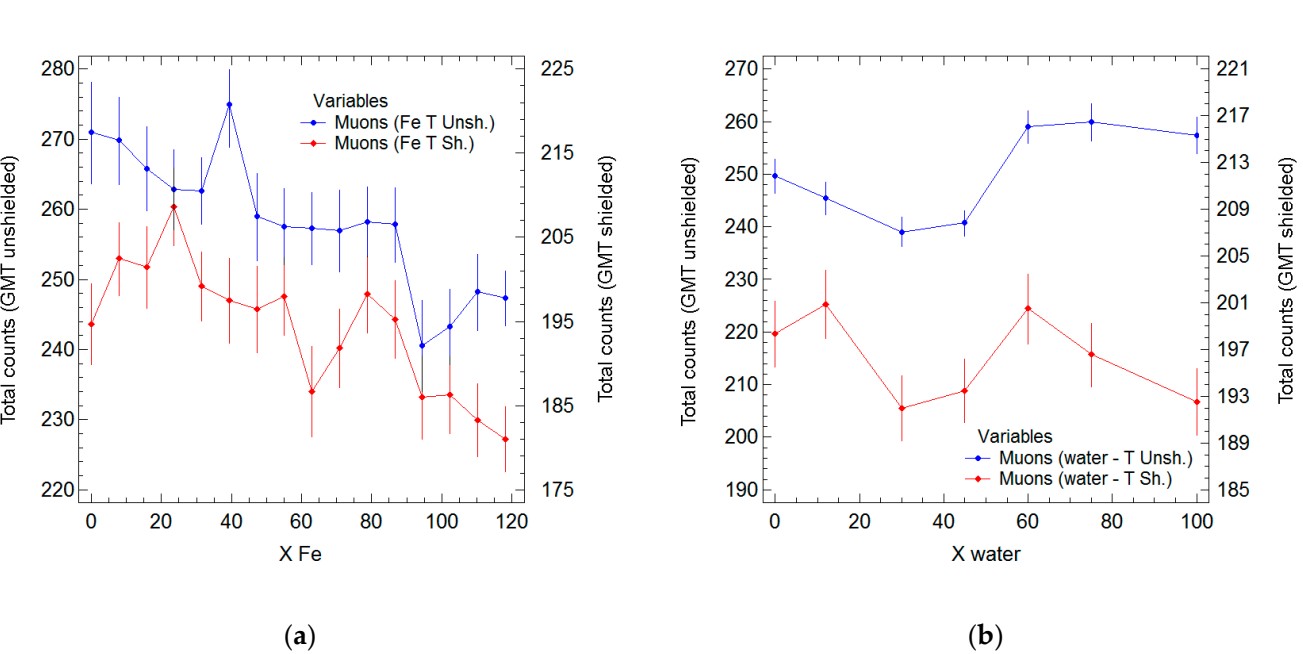

(**a**) (**b**)

**Figure 12.** Measurement of muons in water and iron; the value on the abscise is expressed in radiation depth (gcm$^{-2}$): (**a**) Comparison between two sets of data from iron, shielded vs. unshielded GMTs (total events); (**b**) Comparison between two sets of data from water, shielded vs. unshielded GMTs (total events).

In the case of iron, with increasing thickness, the counting of muons progressively drops, as expected in this type of experiment. The trend could be roughly fitted with some exponential function similar to Equations (1)–(5); the results from an analysis of the linear trend are shown in Table 4.

**Table 4.** Results from a quick simple regression analysis; possible coefficients (least squares) for modeling the results of the experiment of muon absorption in iron.

| Experiment | Intercept | Slope | R$^2$ (%) | Std. Err. of Estimate | Mean Abs. Err. | Model |
|---|---|---|---|---|---|---|
| Iron GMTs unshielded | 5.6 | −0.00087 | 75 | 0.020 | 0.013 | $e^{(5.6-0.00087 \times Fe)}$ |
| Iron GMTs shielded | 5.3 | −0.00084 | 64 | 0.025 | 0.017 | $e^{(5.3-0.00084 \times Fe)}$ |

With water, the results are quite surprising; apparently, from 0 to 45 $gcm^{-2}$, muons are absorbed, but then the counting increases again. This may have some relation to the inflexion visible around 50 $gcm^{-2}$ in Figure 10a and around 30 $gcm^{-2}$ in Figure 10b. Over 60 $gcm^{-2}$, in the case of GMTs shielded, the muon counting abruptly drops again; instead, without the shielding, the counting diminishes slightly.

## 5. Discussion

Considering the simplicity of the instrument, the results are quite interesting. The only issue concerns the interpretations about the background of the detector during the shower measurements. As said in paragraph 4, we should have almost zero accidental coincidences (0.045 cph, or $1.26 \cdot 10^{-5}$ counts per second), so the non-zero value recorded means that from time to time, there are shower detections even without material above the detector (apart from the building roof). The data logger records the number of events (showers and muons) every minute (cpm). We do not have a time stamp for every event, but we noticed that a shower event is present if and only if some muon events are also present. There are at least three possible explanations for this behavior:

1. The simpler explanation is that muons belonging to the same atmospheric shower pass through the water and simultaneously hit all the GMTs. It would be extremely difficult to avoid this possibility, even with a more complex instrument setup and anti-coincidence circuits;
2. Another possible explanation is that sometimes an energetic muon can produce a shower directly inside the instrument by scattering tertiary particles from the metallic surface of the sensors, leading to the counting of a shower event;
3. The phenomenon could result from a combination of the above factors, and local natural radioactivity.

In these experiments, to reduce the background radiation, lead is often used to shield the sensors. Even though we forecasted this behavior, placing the lead shields did not resolve the problem completely, and the shower background still remains. This is a clue for a future improvement of the instrument, maybe with thicker shields or a greater distance among GMTs, or even by using less sensitive sensors.

About the second big maximum detected in the experiment with iron, there are no modern experiments of this type to compare the data; however, this could be a real effect recorded exactly because of the high sensitivity of the GMTs. Anyway, in order to have more precise results, we need better statistics; this will be a possible target for future investigations, as well as to assess the weird results of muon detection in water.

The most important result is surely the proof that cosmic rays in water produce electromagnetic cascades. This, of course, is theoretically expected, but it is the first time that we see this effect with our eyes and by using an old-style device. An interesting result is also the evidence that the showers in water are produced at a greater depth than in metals, and the energy of a particle to initiate a cascade in the water must be higher than in metals. The particle that starts a cascade can be an electron or a photon, but even a muon can initiate a shower by bremsstrahlung or pair production (see Equation (7)). With this instrument, it is not possible to distinguish the culprit; indeed, it is a hard task even for more complicated devices.

The observation of electromagnetic cascades resulting from secondary cosmic ray interactions in water has significant implications for astrobiology. These types of experiments emphasize the potential role of high-energy ionizing radiation in the origin and the evolution of life on Earth and possibly on other planets. Researchers have long speculated that high-energy particles, such as cosmic rays or solar energetic particle events (SEP), could have played a vital role in the origin of life on Earth. All cosmic particles have the ability to induce chemical reactions and ionize molecules, which can lead to the formation of complex organic compounds, including those essential for life, e.g., [17–19]. We hope that our instrument along with its experiments can be of inspiration for students and young researchers, encouraging further exploration in this field, maybe to test the idea that cosmic

radiation may have played a role in the formation of biologically relevant molecules in the early water basins of Earth. Water is definitely an essential element for the formation of life in the universe, and in laboratory experiments, it has frequently shown unique features, and perhaps it will hide other surprises for us in the future.

**Author Contributions:** Conceptualization, M.A.; methodology, M.A.; software, M.A.; formal analysis, M.A.; investigation, M.A., D.L. and A.G.; data curation, M.A.; writing—original draft preparation, M.A.; writing—review and editing, M.A., D.L. and A.G.; visualization, M.A., D.L. and A.G.; project administration, M.A. All authors have read and agreed to the published version of the manuscript.

**Funding:** This research received no external funding.

**Data Availability Statement:** The data presented in this study are available on request from the corresponding author. The data are not publicly available due to proprietary rights or intellectual property considerations, as they may be part of ongoing research.

**Acknowledgments:** We acknowledge all the people and institutions supporting the ADA project.

**Conflicts of Interest:** The authors declare no conflict of interest.

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
