# Peer review of "Exploring the Interaction of Cosmic Rays with Water by Using an Old-Style Detector and Rossi’s Method"

_2571-712X, doi:10.3390/particles6030051_

Round 1
Reviewer 1 Report
I have concerns over the content and method of this paper on several counts.
Firstly, with regard to the Abstract (and onwards through the text), the distinction between primary cosmic rays and the secondaries they generate should be clarified so it's made clear exactly what particle detection is being discused in relation to the iron or water-based GMT devices. On Earth, cosmic rays are almost invariably stopped in the atmosphere and the primary particles very rarely reach the ground. Only the secondaries from subsequent air showers will do so. Whether primary or secondary particles, both are less likely to be stopped by a water-based detector than an iron-based one owing to the higher Z-value of the latter, and as to whether primary or secondary particles will interact in water, in my scientific opinion this is not in doubt. We know that primary cosmic rays interact in air, so due to their similar constituent components (33.1% of the atoms in water are oxygen, compared to ~100% of oxygen and nitrogen - with similar Z - in air) and the much higher density of water, of 1.025 g/cm2, it is almost inconceivable that cosmic rays, and their secondaries, will not interact in water.
Virtually all particles detected at ground level in either a water or iron based detector will be secondaries such as muons, and not primary cosmic rays. It should be made clear that the experiment will be measuring the interaction, not of primary cosmic rays, but of the secondary particles from the subsequent ait shower, with water (or iron).
In addition to these concerns, the use of water to detect cosmic ray air showers is not remotely new, and detectors such as HAWC and LHAASO use water as a Cherenkov radiator to do exactly this, with detection of muons used to veto hadronic events so as to focus on very high energy photon-produced showers for ground based gamma-ray astronomy. This closely related technique is not discussed at all but should at least be mentioned. It is also likely that the particle cascade continues within the large Cherenkov water tanks employed in these observatories - the effect that is being measured is these experiments.
In terms of a simple experiment to demonstrate the presence cosmic rays (or more accurately the secondaries produced by these), the Cherenkov light produced in the water tank is easier to detect with readily available PMTs than shower secondaries using GMTs, which are rare devices these days.
Treatment of errors is extremely important in an experiment such as this, especially given the relatively small size of the datasets. I think it is very important that an analysis of errors is undertaken. This would provide some quantitative measures of the accuarcy of the data on order to indicate e.g. whether the difference between energies required for iron and water for shower initiation are different as suggested, and whether the presence of the “the maximum of the shower transition curve” is a valid proposition .
A separate file containing suggested corrections and amendements to the report text is attached.

The quality of the English is generally good. I have noted issues that would benefit from modification in the attached file.
Author Response
Dear reviewer,
thank you for your time and suggestions. We have made all the possible required modifications to the paper; in the attachment is the point-by-point response.

Reviewer 2 Report
In this work, the authors designed an old experiment to measure the electromagnetic cascade in water. First, he reviewed the theoretical knowledge of the cascade with a lot of space, and then obtained the evidence of Cosmic ray electromagnetic cascade in water through experiments. And I have some questions as following:
1. In Figure 8, there are difference between the results with GMT unshielded or shielded, why?
2. In Figure 9, the results is in doubt, the number of cascades in distilled water much higher than that in tap water by about 20%? Need to explain clearly.
3. In Figure 10, Figure 10 (a), why did the fourth point fall off? Figure 10 (b) shows two platforms, explained by the second platform, but what is the reason for the first platform? Need to explain clearly
4. 3.1. Several GMT detectors are described in confusion, please check.
The wring need be polished.
Author Response

(The authors gave the same response as above.)

Round 2
Reviewer 1 Report
The authors have largely implemented the suggested changes, and I am happy now to recommend that this paper is publishable, apart from the minor revisions noted below.
Line 242 : I suggest this is changed to "found to be unity"
Figure 5 caption: I suggest this is changed to "Simplified block diagram .."
Given the magnitudes of the errors reported, the number of decimal places given in tables 2 and 4 are too high.
Line 588: I suggest this is changed to "need better statistics .. "
I am happy now to recommend that this paper is publishable, apart from the minor revisions noted below.
Author Response
All the modifications have been done, and the tables are now more "clean". We have highlighted the corrections in yellow as required by the editor.
Thank you again for your time and patience.
Reviewer 2 Report
The paper can be published in current revised version.
Author Response
Thank you again; we've very much appreciated your time and patience.